# OpenReview forum: "Transferable Adversarial Attack on Vision-enabled Large Language Models"
_ICLR.cc/2025/Conference — ICLR 2025 Conference Withdrawn Submission_

### Official Review · Reviewer_L8rG · 2024-10-22

**Soundness:** 3
**Presentation:** 2
**Contribution:** 3
**Rating:** 5
**Confidence:** 4

**Summary:**

The paper proposes two techniques to create adversarial attacks to Vision-enabled Large Language Models (VLLMs).

1) CLIP Score attack - this technique essentially learns a perturbation to an image such that the embedding of the image aligns with (high cosine similarity to) a set of embeddings of incorrect text labels, and does not align with a set of embeddings of correct text labels.
2) VLLM response attack - this technique learns a perturbation to directly maximizes the log likelihood of the VLLM outputting some incorrect label when prompted with an image and text query.

Both techniques learn perturbations using gradients from white-box surrogate models. The paper focuses on how the learnt adversarial attacks transfer to help out models (black-box transfer).

The paper tests the above techniques in three different settings:

1) Image classification
2) Text generation (captioning of images in natural language)
3) Safety-related reasoning (identifying properties of harmful images or answering safety-related questions about harmful images).

In each setting the paper demonstrates impressive transfer of attacks to held out models. Most impressively, across all settings they create successful attacks (non 0 attack success rate) to frontier models such as GPT-4o and Claude 3.5 sonnet.

In addition to these results, the author's create a custom dataset and benchmark for the "Safety-related reasoning" task and release this.

**Strengths:**

From here on, I will refer to changing the data present in an image as an adversarial attack to a VLLM. That is I will not be including things such as jailbreaks or prompt injection attacks in my notion of adversarial attacks.

### Originality

Whilst adversarial attacks to VLLMs have been studied widely, the authors make the following original (to the best of my knowledge) contributions (in order of importance):

1) Most importantly, they show the highest attack success rate for transferable adversarial attacks I have seen.
2) Through ablation studies, they show how various "tricks" such as data augmentation and model ensemble size can be used to enhance transferability. By "tricks" I do not mean anything negative, and believe that these techniques are useful for the broader research community to know about.
3) They introduce three different evaluation tasks for VLLMs.
4) They release a dataset and benchmark for the third of said tasks, safety-related reasoning.

### Quality and Clarity

The paper is high quality. The experiments are thorough and I am convinced of the broad claims made in the paper. For the most part the paper is well written and easy to follow, and figures / tables are informative.

### Significance

In my opinion, adversarial attack papers are most significant when they demonstrate methods that can be deployed against, or at least one can imagine a scenario in which they would be deployed against, real world systems. In the case of attacking frontier models, this means a technique is significant if it can be used against close-source models such as GPT4, Claude, etc.

This paper meets this criteria.

**Weaknesses:**

I am going to split my critique up into two sections. The first will be a high-level critique of the paper, and the second will be specifics about sections. Whilst the critique is long, this is only because I believe the paper has interesting results that could be improved, not because I think there are any fundamental failings in the paper. To the contrary, I think the paper contains valuable insights for the broader adversarial attack community.


# High Level:

### Originality

My high-level critique of this work concerns its originality. In particular, the CLIP Score attack and VLLM response attack seem very similar to the two attacks presented in Dong et al.[1]. In particular, Dong et al. also present a method based on CLIP embeddings (albeit they align to a target image not a target textual embedding) and an end to end technique. They also demonstrate that these methods transfer to black-box models. Zhao et al. [2] also demonstrate black-box adversarial attacks of a similar nature to those presented in this work (although they do not attack frontier models).

Firstly, these works should be mentioned and treated in the related works, but they are not ([1] is however mentioned at the end of the introduction).

Despite this critique, I believe this paper still has valuable contributions. In particular, making a technique successful in machine learning often comes down to small tweaks or "tricks". The authors demonstrate, that through the specific techniques they use, they are able to get what appears to be stronger transfer than [1].

I would recommend two concrete changes on this front however:
1) Soften or remove claims of novelty about your techniques. For example, you state "we develop a novel attack for VLLMs designed to find image perturbations by targeting adversarially chosen text embeddings." Given what I have seen, I do not think it is fair to say your technique is entirely novel. An alternative framing would be to say something like "building on prior works that have displayed some transferability <cite>, we enhance transferability by doing <x>".
2)  Running baselines using prior works ([1] in particular) and seeing how adding your tricks (ensembling, data augmentation) would be very valuable.

### Experiments

 In all of the experiments I do not understand the exact algorithm you are using. You say you use an ensemble of surrogate models that includes CLIP models and full VLLMs. This leads me to assume that the adversaries are created using a mixture of the CLIP score attack and VLLM response attack methods (e.g. you accumulate loss from both and then take a gradient step). Is this correct?

 If this is correct, then why did you not show ablation studies of using each technique individually? This would seem to me to be very important. If you find that using both techniques at the same time was what increased transferability, that would be a very useful to know. Additionally, if this is correct or not, language should be added to make the experimental setup more clear.

Apologies in advance if I have missed something here.


### Definition of adversarial attack

Secondly, the paper uses a broad definition of adversarial attacks to VLLMs. For example, in the related works you compare to Schaeffer et al. Their paper concerns transferable jailbreaking attacks. In contrast this paper concerns adversarial attacks that change how the model perceives an image, as opposed to attacks that convey some hidden instruction to the model. In fact, your formulation in equation (1) is good (although I have some critiques of it below) and clearly does not cover the case of jailbreaking or prompt injection attacks. Making this distinction clearer in the introduction and related works would be valuable,

# Section level critique

### Section 2 - related works

- You state "in this paper, we take a new perspective: We
	investigate how visual perturbations can induce targeted misinterpretations in proprietary VLLMs
	such as GPT-4o." As mentioned above, I don' think this is an entirely new perspective, and thus should be removed.
- Like I said above, a more thorough treatment of [1] and [2] is required.

### Section 3 - generating transferable attacks for VLLMs

- The problem setup in equation (1) seems slightly off. The requirement is only that the two outputs are not equal, but this would be satisfied by simply flipping a single token (which does not match my internal definition of what I think an adversarial attack to be). For example, this could occur simply if I was sampling with some entropy.
- Nit - above equation (5) I think tilda x_a should be tilda t_a? That is tilda t_a is introduced in (5) without a definition.

### Section 4 - experiments

- See above concerning my question of method.
- For each experiment, I would like to see for each epsilon budget how a random perturbation of that size affects the model performance. It may be in all cases that the ASR remains 0, in which case I think this is still valuable to include but can be put in the Appendix. If not, then this is a useful baseline to compare your method against.
- For table 5, it would be nice to see some example questions, model responses with and without adversarial perturbations.
- Nit - should be "after generating" not "after generate" in line 302.
- Section 4.3. Claims here need to made more narrow. You state "This benchmark not only provides a standardized framework for assessing VLLM capabilities in safety critical domains." This is evaluating a certain subset of safety critical domains. Not all possible VLLM safety critical domains (e.g. it does not tell me much about how useful the VLLM could be used as an agent to assist me in some nefarious task). This is ok, just the claim should be made more narrow.
- I think the safety benchmark is very interesting. Writing could be enhanced by referencing real world situations in which a failure on this benchmark would lead to bad things.

### Other Nits

- Line 335 "thereby making transferable attacks more less effective on Claude’s models", should be "more OR less"
- Line 392 "Each question ends with 'Please
	answer with yes or no'"
		- The quotation mark is backwards.

References:

[1] - Dong, Yinpeng, et al. "How Robust is Google's Bard to Adversarial Image Attacks?." arXiv preprint arXiv:2309.11751 (2023).
[2] - Zhao, Yunqing, et al. "On evaluating adversarial robustness of large vision-language models." Advances in Neural Information Processing Systems 36 (2024).

**Questions:**

My questions simply relate to the weaknesses raised. I restate them in more brevity here:

Q1) Originality. Do you believe my concerns with regards to originality are valid, and if so how do you intend to edit the paper accordingly?

Q2) Experiments. Is my interpretation of the method used to produce the results correct (a mixture of the two attacks presented), and if so why did you not compare the techniques individually?

Q3) Definition of adversarial attack. Do you agree with my distinction between the types of adversarial attack, and if so how do you intend to edit the paper to reflect this?

Q4) How do you plan to address the other more narrow weaknesses I raised about each of the sections?

Overall I think this is a valuable piece of work! I believe, however, that it could be made stronger by adressing these concerns.

---

### Official Review · Reviewer_jHNg · 2024-10-28

**Soundness:** 2
**Presentation:** 2
**Contribution:** 2
**Rating:** 3
**Confidence:** 4

**Summary:**

This paper explores the vulnerability of vision-enabled large language models (VLLMs) to adversarial attacks, focusing on the transferability and universality of adversarial examples. The authors introduce two specific attack methods—CLIP Score and VLLM Response attacks by attacking the vision modality of the VLLM—demonstrating their impact across two tasks: image classification and text generation and six VLLMs.

**Strengths:**

This paper is both relevant and timely, given the real-world deployment of models with these vulnerabilities and the associated risks. The authors conduct thorough experiments on these models, providing an in-depth assessment of their adversarial robustness. The discussion of universal perturbations for VLLMs is especially compelling, as it addresses an area that remains underexplored.

**Weaknesses:**

My main concern is that the proposed attack lacks novelty as in [1], the authors introduced a transfer-based attack strategy by matching image-text features. Many of the conclusions drawn like the vulnerability of VLMs to adversarial attacks-specifically on the vision modality of these models- and the transferability of these attacks are already well-explored in the existing literature, which the authors fail to acknowledge adequately [1,2].
Throughout this paper, there is a notable lack of consistency and cohesion across sections. Furthermore, the VLLM SafeBench tool is introduced without any prior context, emerging only within the experimental results. The experimental design is disorganized, with unclear settings. Overall, the paper lacks clarity, cohesion, and originality.


1.	Zhao, Yunqing, et al. "On evaluating adversarial robustness of large vision-language models."
2.	Yin, Ziyi, et al. "Vlattack: Multimodal adversarial attacks on vision-language tasks via pre-trained models."

**Questions:**

Below is a list of comments and questions, ordered not by importance:

- Lines 121-122: Why are you comparing the success rate of untargeted and targeted attacks?
- Line 180: How do you know this? Did you conduct experiments to verify?
- The threat model for the two proposed attacks is unclear. Specifically, what level of access does the attacker have? Is this a white-box or black-box attack, and is it targeted or untargeted?
- Line 186: The definitions of "transferable" and "universal" should not be relegated to the appendix.
- I do not see the novelty in the CLIP Score attack. In [1], the authors introduced a transfer-based attack strategy by matching image-text features. What is your contribution beyond this approach?
- Why are you proposing two separate attacks? The motivation presented in lines 218-219 is unclear and insufficiently supported. From my understanding, the CLIP Score attack is untargeted, while the VLLM Response attack is targeted.
- In line 225, I am assuming  $\tilde{t_a}$ should be $\tilde{x_a}$? In general, the optimization for the VLLM Response attack would benefit from more explanation, what do you mean by its output? is it the target text or a label?
- Which attack are you referring to as the "transfer attack" in line 268?
- In Section 4.1, the experimental setup is unclear. The surrogate and victim models are not identified in the tables, and it’s not specified which attack is being optimized.
- In lines 335-336, you suggest that the attack is only transferable when the same or a similar model is used for embeddings. Is this correct?
- The term "Multimodal LLMs" is introduced only in certain section titles and table captions.

---

### Official Review · Reviewer_ufji · 2024-11-04

**Soundness:** 2
**Presentation:** 3
**Contribution:** 2
**Rating:** 3
**Confidence:** 4

**Summary:**

This paper presents an adversarial attack to demonstrate that the targeted adversarial examples are transferable to current VLLMs. By crafting perturbations, the attacker can achieve both good targeted and untargeted attack. The author also finds that universal perturbations can consistently induce the misinterpretations across VLLMs.

**Strengths:**

1.The paper is well-written and easy to follow.

2.The selection of Open-sourced VLLMs are SOTA.

3.Investigating the attack on VLLMs is necessary for the VLLM safety community.

**Weaknesses:**

1. Provide some adversarial examples (visually)? If we just randomly perturbed the images, it can also cause the misclassification. So it can not prove the efficiency of the proposed method.

2. The paper use ε=32/255 (%) to measure the perturbation. If the perturbation is too much, it is not stealthy. we can see that when the 32/255, it can get some good results. But this perturbation is a bit large and it will affect the stealthies.

3. The paper claims to achieves good attack performance on both untargeted and targeted attack settings. For targeted attack setting, the results are only provided by ImageNet-1K image classification. More experiments would be better.

4. The proposed two attack methods: CLIP score attack and VLLM response attack, are good, however not novel. The intuition has been proposed by other works, and the author applied to the adversarial attack domain.

**Questions:**

1.Seems the attack only manipulates the image. Is it possible to add some analysis on the attack effects on text perturbation?

---

### Official Review · Reviewer_ojxe · 2024-11-05

**Soundness:** 2
**Presentation:** 2
**Contribution:** 3
**Rating:** 5
**Confidence:** 4

**Summary:**

The paper explores whether targeted adversarial examples are transferable to widely-used proprietary Vision-enabled Large Language Models (VLLMs) such as GPT-4o, Claude and Gemini. The paper conducts experiments to show that crafted perturbations by attackers can induce the misinterpretation of visual information. Also, the paper shows that universal perturbations can consistently induce these misinterpretations. The paper conducts sufficient experiments including object recognition, visual question answering and image captioning.

**Strengths:**

The paper explores the transferability of adversarial examples to proprietary blackbox Vision-enabled Large Language Models (VLLMs). This transferability of adversarial examples is significant.

Two attacks including CLIP score attack and VLLM response attack are proposed. Also, two tricks including data augmentation and ensembling surrogate are proposed to enhance transferability.

The paper conducts sufficient experiments including using two open-source VLLMs, using three blackbox VLLMs, and conducting attacks on three tasks.

**Weaknesses:**

The paper only provides the experiments results to demonstrate the transferability of proposed two attacks and tricks. It is better to explain more about the reason of transferability.

It looks like that proposed attacks depend on the positive and negative textual prompts in equation (2). It is better to do some explainations or experiments to show the influence of textual prompts.

There are some typos, e.g., the confusing use of t_a^~ and x_a^~ in line 223 and equation (5).

There are no comparison methods in the main results. It is difficult to understand the advantage of proposed attacks.

**Questions:**

Since some works have explored the transferability of adversarial attacks, could the authors explain the difference of transferability between traditional adversarial attacks and proposed attacks?

---

### Note · Authors · 2024-11-25

I have read and agree with the venue's withdrawal policy on behalf of myself and my co-authors.